# Thermodynamic Characterization of Rhamnolipid, Triton X-165 and Ethanol as well as Their Mixture Behaviour at the Water-Air Interface

**DOI:** 10.3390/molecules28134987

**Published:** 2023-06-25

**Authors:** Anna Zdziennicka, Maria Luisa González-Martín, Edyta Rekiel, Katarzyna Szymczyk, Wojciech Zdziennicki, Bronisław Jańczuk

**Affiliations:** 1Department of Interfacial Phenomena, Institute of Chemical Sciences, Faculty of Chemistry, Maria Curie-Skłodowska University in Lublin, Maria Curie-Skłodowska Sq. 3, 20-031 Lublin, Poland; anna.zdziennicka@mail.umcs.pl (A.Z.); rekieel@gmail.com (E.R.); katarzyna.szymczyk@mail.umcs.pl (K.S.); bronislaw.janczuk@mail.umcs.pl (B.J.); 2Department of Applied Physics, University Institute of Extremadura Sanity Research (INUBE), Extremadura University, Avda. de Elvas, s/n, 06006 Badajoz, Spain; 3Networking Research Center on Bioengineering, Biomaterials and Nanomedicine (CIBER-BBN), 06006 Badajoz, Spain; 4University Clinical Hospital in Poznań, Przybyszewskiego 49, 60-355 Poznań, Poland; wojtekzdziennicki@gmail.com

**Keywords:** rhamnolipid, ethanol, Triton X-165, mixture, adsorption

## Abstract

In many industrial fields, in medicine or pharmacy, there are used multi-component mixtures of surfactants as well as more and more often mixtures containing biosurfactants. Thus, in our study the mixtures of rhamnolipid (RL), ethanol (ET) and Triton X-165 (TX165) were applied. For these mixtures the surface tension of aqueous solutions with constant concentration and composition of ET and RL as well as the variable concentration of TX165 was measured. Based on the obtained results and the literature data, thermodynamic analyses of the adsorption process of ET, RL, TX165, binary mixtures of ET + RL, ET + TX165 and RL + TX165 as well as the ternary mixtures of RL + ET + TX165 at the water-air interface were made. This analysis allows to propose a new equation for calculation of the total ethanol concentration at the water-air interface using the Guggenheim-Adam adsorption isotherm. The constants in the Langmuir and Szyszkowski equations for each component of the studied mixtures as well as the composition of the mixed monolayer at the water-air interface were also successfully analysed based on the contribution of particular surface active compounds to the water surface tension reduction as well as based on the Frumkin isotherm of adsorption.

## 1. Introduction

The use of surfactants in various post-industrial branches as well as in everyday life is so important that it is difficult to imagine modern economy without them. It should be remembered that classic surfactants are the products of chemical synthesis [1]. Despite the fact that they are not a direct threat to human life they can cause allergies. Additionally, surfactants discharged in sewages can penetrate into ground and surface waters accumulating in our environment [2,3]. The accumulation of surfactants in large quantities and their low degradation become a real threat to fauna and flora. In addition, the surfactants adsorption enables their penetration to the cell membranes and causes pathological changes in living organisms. As a result, more and more attention is paid to the protection of our environment, and thus some attempts are made, among others, to eliminate harmful compounds. For this reason, the research aimed at developing new compounds characterized by complete biodegradation in the environment. Moreover, their obtainment from raw materials and renewable sources or waste products is of significant importance. These types of compounds include biosurfactants, which become more and more interesting [4,5,6,7,8]. Unfortunately, the high cost of their production is one of the important barriers to their widespread use in practice. One of the ways of the practical application of biosurfactants, despite the high cost of their production, is to mix them with classic surfactants, such as Tritons which are still find in practical application [9,10]. 

Among biosurfactants, rhamnolipid (RL) is special importance and among Tritons it is Triton X-165 (TX165). Rhamnolipid is characterized by large surface activity, low CMC and a number of biological activities such as antiviral, anticancer as well as preventing the biofilms formation [11,12,13,14,15,16,17,18,19,20]. In fact, rhamnolipid is a mixture of up to 26 different compounds [21]. Therefore one can find differ opinions on its adsorption and aggregation activity in the literature [22,23]. It should be noted that most of the available papers deal with mono-rhamnolipid. Besides the above mentioned properties rhamnolipids characterized by the antiviral activity against herpes simplex viruses and HSV. It is effective in the treatment and alleviation of psoriasis, chronic wounds, including burns and advantageous minimizing scarring. According to the recent studies rhamnolipid can be applied in oncology [20]. 

In turn, Triton X-165, like other surfactants in this group, is neutral, non-toxic as well as compatible with anionic and cationic surfactants [24]. Compared to the ionic surfactants, the nonionic nature of Tritons is considered physiologically inert. For this reason and their low CMC, they are considered the safest drug delivery compounds regardless of dilution in the human body [25]. 

Taking into account such characteristic functional properties of rhamnolipid and Triton X-165, their mutual influence on the adsorption and aggregation activity as well as the related composition of micelles and the surface layer at the water-air interface was examined [26]. It was stated, among others, that the composition of the mixed monolayer at the water-air interface can be deduced based on the surface tension isotherms of individual RL and TX165 and that there is a synergetic effect in the reduction of the water surface tension by the mixed monolayer and in the micelles formation at a given composition of the RL and TX165 mixtures. 

As it is commonly known the positive effects of the mutual influence of RL and ethanol (ET) on their adsorption properties is observed which can be insufficient in some practical applications. It should be added that ethanol has been known as a disinfectant for many years, and the coronavirus pandemic has revived that interest. Thus, the mixture of RL + TX165 with ET can be interesting not only from the theoretical point of view of the adsorption process of multi-component mixtures at the water-air interface, but also from its practical application. 

As a matter of fact, one can find in the literature many papers dealing with the adsorption of ethanol mixtures with various surfactants at the water-air interface [27,28,29,30]. However, there are no papers in which the process of adsorption at this interface of the mixture of the biosurfactant with the classical surfactant in the presence of ethanol is considered. The thermodynamic interpretation of the adsorption process of surfactants and ethanol at the water-air interface in the whole range of their concentrations is difficult, among others, because the Gibbs surface excess concentration of ethanol is not equal to its total concentration in the mixed monolayer and that in both surface region and the bulk phase ethanol and water can be treated as a mixture but not as a solution. The difference in the interpretation of the surface behaviour of ethanol at the water-air interface not only in the mixture with surfactants but also in their absence may be result of the above. Therefore the aim of the our studies was the thermodynamic characterization of the RL + ET + TX165 mixtures behaviour at the water-air interface based on the surface tension data of the aqueous solution of these mixtures at the constant RL + ET concentration at different compositions and variable concentrations of TX165.

## 2. Results and Dissuasion

### 2.1. Some Physicochemical Properties of Water, RL, ET and TX165

To understand the behaviour of the ternary mixture of RL, ET and TX165 at the water-air interface, some of its and water physicochemical properties must be known.

The analysis of the bond length between O and H, the angle between the -OH bonds and the average distance between the water molecules indicates that a water molecule at the temperature of 293 K can be inscribed in a regular cube with the edge length of 3.11 Å. Thus, the contactable area of water molecules with other molecules is 9.7 Å^2^. This value is close to that of the contact area determined by Groszek [31] based on the water vapour adsorption on the quartz surface, which is equal to 10 Å^2^. This means that 1 × 16.6^−^^6^ mole of water is needed to cover the given surface by its monolayer. The value of the water molecule contactable area equal to 10 Å^2^ is very often used for the thermodynamic consideration of the surfactants adsorption at the water-air interface [26,32]. This adsorption decreases the water surface tension. According to van Oss et al. [33,34,35] water is the bipolar liquid and its surface tension results from the Lifshitz-van der Waals and Lewis acid-base intermolecular interactions. The Lewis acid-base interactions lead to the hydrogen bonds formation between the water molecules. Thus, the water surface tension can be divided into the Lifshitz-van der Waals component (LW) and the acid-base (AB) one, which results from the electron-acceptor and electron-donor parameters. As a result of the surfactants adsorption at the water-air interface, the surface tension of water is reduced, especially its AB component [26,32]. 

It proved that the adhesion work of the aqueous solution of many hydrocarbon surfactants to the PTFE surface does not depend on their concentration and is equal to the water adhesion work [36,37]. From the Young-Dupre equation and Fowkes approach to the interface tension [38], it follows that in this case the LW component of the surfactants solution and water is the same and its value is equal to 26.85 mN/m at 293 K (Table 1). This value is considerably higher compared with that determined based on the water-*n*-alkane interface tension [38,39]. As a matter of fact, for water it was assumed that the electron-acceptor and electron-donor parameters of the acid-base component of its surface tension are the same [33,34,35]. It should be remembered that the parameters of other liquids or solids surface tension are consistent with this assumption. These parameters for the ethanol surface tension ($\gamma_{LV})$ are not the same (Table 1). However, the LW component of the ethanol surface tension is close to that of water determined based on the water-*n*-alkane interface tension [29,30,39].

The ET molecule volume calculated using the bonds length and the angle between them as well as the average distance of molecules in the bulk phase at 293 K is close to 97 Å^3^ and to the value obtained from the ET density, which is equal to 97.3 Å^3^. As follows from the calculations the ethanol molecule can be put in a regular cube with the edge equal to 4.6 Å [30]. Thus, the contact area of the ET molecule with other one does not depend on its orientation and is about 21 Å^2^ [30]. This point out that one ET molecule can replace two water molecules in the interface monolayer. 

Unlike ET and water, the surface tension of RL and TX165 depends on the orientation of their molecules towards the air phase (Table 1). If their molecules are oriented with the hydrophobic group towards the air phase, then we treat the surface tension as the surface tension of the tail. However, with the orientation of the RL and TX165 molecules with the hydrophilic group towards the air, this tension is called the surface tension of the head (Table 1).

The tail surface tension of TX165 and RL is close to LW of ET and water determined from the water-*n*-alkane interface tension. However, these values are considerably smaller than LW for water determined from the contact angle of water on the hydrophobic solids [40]. Based on the analysis of the chemical bonds length, the angle between them and the average distance between the molecules at 293 K, it appears that the RL and TX165 molecules cannot be entered into one cube, but into different ones, for the head and tails of their molecules, respectively [26]. It follows from this analysis that the contactable area of RL and TX165 molecules at their perpendicular orientation towards the interface is equal to 69.09 Å^2^ and 35.7 Å^2^, respectively. At the parallel orientation of RL molecule towards the air phase the contactable area of its tail is equal to 87.3 Å^2^ and that of the head to 72.1 Å^2^ [26]. In the case of TX165 at the parallel orientation of its molecule at the interface, the contactable area of the tail is equal to 52.12 Å^2^ and that of the head to 101.4 Å^2^. Taking into account the contactable area of water and ET it can be stated that in the bulk phase one molecule of ET can be surrounded by 12 molecules of water. The tail of the TX165 can be surrounded by about 20 water molecules, and the head can be bound by strong hydrogen bonds with about 40 molecules and weak ones also with 40 water molecules. In the case of RL its tail can be surrounded by about 30 molecules of water and the head by 28 ones.

The number of the water molecules contacted with the tail and the head of surfactant decide about its tendency to adsorb at the water-air interface. In turn the monolayer formed at the water-air interface reduces the water surface tension. In fact, in the case of ET the water surface tension is reduced to that of ET because the ET is infinitely miscible with water (Appendix A). One would expect that the minimum surface tension of RL and ET aqueous solutions should be close to the surface tension of their tails. In the case of RL, the minimum surface tension of its aqueous solution is not much higher than LW for the water obtained from the contact angle on the surface of hydrophobic solids. However, in the case of TX165 there is a great difference between the minimal surface tension of its solution and LW for water (Appendix A) (Table 1). This may be due to the fact that the number of water molecules surrounding the head of the TX165 molecule are much greater than that of water molecules surrounding its tail.

The isotherms of the aqueous solution of ET, RL and TX165 can be described by the exponential function of the second order which has the form:(1)$$\gamma_{LV}=y_{0}+A_{1}\exp\left(\frac{-C}{t_{1}}\right)+A_{2}exp\left(\frac{-C}{t_{2}}\right)$$
where $y_{0}$, $A_{1}$, $A_{2}$, $t_{1}$ and $t_{2}$ are the constants, C is the concentration. 

These constants are probably connected with particular intermolecular interactions between the water molecules and the surface active ones. It was earlier stated that the $y_{0}$ constant is related to the LW intermolecular interactions and the other constants to the acid-base ones.

The possibility to describe the isotherm of the surface tension is very useful for determination of the surface Gibbs excess concentration (Γ). The surface tension isotherms of the aqueous solution of RL, ET and TX165 can be also described by the Szyszkowski equation which has the form [1]:(2)$$\gamma_{0}-\gamma_{LV}=RT\Gamma^{max}ln\left(\frac{C}{a}+1\right),$$
where $\gamma_{0}$ is the solvent surface tension, *R* is the gas constant, *T* is the absolute temperature, $\Gamma^{max}$ is the maximal Gibbs surface excess concentration and a is the constant. 

This equation can be applied for determination of the constant a related to the maximal Gibbs surface excess concentration. Additionally, the surface tension isotherm of the aqueous solution of ET can be described by the Connors equation [30,41], which is as follows:(3)$$\gamma_{LV}=\gamma_{0}-\left(\gamma_{0}-\gamma_{S}\right)\left[1+\frac{\beta \left(1-X_{ET}^{b}\right)}{\alpha \left(1-X_{ET}^{b}\right)}\right]X_{ET}^{b}$$
where $\gamma_{S}$ is the surface tension of alcohol, α and β are the empirical constants and $X_{ET}^{b}$ is the mole fraction of ET in the bulk phase.

In the case of the ET aqueous solution, despite the possibility of describing the surface tension isotherm by the well-defined mathematical function, it is difficult to determine the real Gibbs surface excess concentration at the water-air interface, and more so the total concentration of ET in the surface layer. For this reason one can find conflicting opinions about this issue in the literature [30]. ET is the surface active agent which, as mentioned above, is infinitely miscible with water. Similarly to the classical surfactants, it forms aggregates at a certain concentration in water, however, they cannot be treated as typical micelles [29,30]. Moreover, unlike typical surfactant, at the concentrations above the critical aggregation concentration (CAC), a decrease in the surface tension of the solution is still observed [30]. The number of the ET moles in 1 dm^3^ are also changed as a function of its concentration. Depending on the ET concentration it is treated in the practice as the co-surfactant and/or co-solvent [1,42]. This fact causes that the aqueous solution of ET must be treated thermodynamically in a different way from the aqueous solutions of typical surfactants.

In the case of non-ideal solution in which the solute concentration is small, the chemical potential (μ) of the component of the solution is defined asymmetrically. For the solute it can be written:(4)$$\mu =\mu^{*}+RTlnX^{b}f^{*}$$

In the case of the mixture of solvents the chemical potential is defined symmetrically and can be expressed:(5)$$\mu =\mu^{0}+RTlnX^{b}f^{0}$$
where $\mu^{*}$ and $\mu^{0}$ are the standard chemical potentials, which depend only on temperature and pressure, $RTlnX^{b}f^{*}$ and $RTlnX^{b}f^{0}$ are chemical potentials of mixing, $X^{b}f^{*}=a^{*}$ and $X^{b}f^{0}=a^{0}$ are the activities, $f^{*}$ and $f^{0}$ are the activity coefficients and $X^{b}$ is the mole fraction of the solute.

It is known that in the concentration range from zero to 0.01 mol/dm^3^ most surfactants are present in the bulk phase in the monomeric form which decides about their concentration in the bulk phase [1]. In this range of surfactants concentration it be can assumed that with a small error $X^{b}=\frac{C}{\omega }$ (C is the concentration of surfactants and ω is the number of the water moles in 1 dm^3^) and $f^{*}≅1$. Indeed, in the considered concentration range of the surface active agent ET fulfills such conditions if its chemical potential is defined asymmetrically. In such case it can written:(6)$$\Gamma =\frac{X^{b}}{nRT}{\left(\frac{\partial \gamma_{LV}}{\partial X^{b}}\right)}_{p,T}=-\frac{C}{nRT}{\left(\frac{\partial \gamma_{LV}}{\partial C}\right)}_{p,T}=-\frac{1}{2.303nRT}{\left(\frac{\partial \gamma_{LV}}{\partial logX^{b}}\right)}_{p,T}\;\;\;\;\;\;\;\;\;\;\;\;\;\;=-\frac{1}{2.303nRT}{\left(\frac{\partial \gamma_{LV}}{\partial logC}\right)}_{p,T}$$
where n is the constant which depends on the type of the surface active agent and for ET is equal to 1. 

The studies by Chodzińska et al. [30] proved that in the ET concentration range from 0 to 0.01 mol/dm^3^, the Γ values calculated using C and $X^{b}$ do not differ much. However, the difference increases as the increasing ET concentration. It was concluded by them that the most reliable values of the Gibbs surface excess concentration of ET at the solution-air interface can be obtained from the following equation [26,32]: (7)$$\Gamma =-\frac{a^{0}}{nRT}{\left(\frac{\partial \gamma_{LV}}{\partial a^{0}}\right)}_{p,T}=-\frac{1}{2.303nRT}{\left(\frac{\partial \gamma_{LV}}{\partial loga^{0}}\right)}_{p,T}$$

Indeed, in the case of ET the Γ values are not equal to its total concentration in the surface layer at the solution-air interface. Moreover, they do not tend to zero as $a^{0}$ tends to 1.

To determine the total concentration of ET at the solution-air interface ($\Gamma^{tot}$) the values of Γ should be recalculated as the Guggenheim-Adam ones ($\Gamma^{GA}$). For this purpose there was applied the following equation [43]:(8)$$X_{W}^{b}\Gamma =\frac{V_{s}}{V_{W}}\Gamma^{GA}$$
where $V_{S}$ is the average molar volume of the ET aqueous solution and can be expressed as:(9)$$V_{s}=V_{W}X_{W}^{b}+V_{A}X_{A}^{b}$$
where $V_{W}$ and $V_{A}$ are the molar volumes of water and ET, respectively.

Taking into account the $\Gamma^{GA}$ values for ET which were going to zero if the ET molar fraction achieved unity it was possible to determine the $\Gamma^{tot}$ using the expression [30]:(10)$$\Gamma^{tot}=\Gamma^{GA}+C\times h$$
where h is the ET molecule length.

It appeared that the $\Gamma^{tot}$ values calculated in this way are not linear from C corresponding to the maximum of $\Gamma^{GA}$ of the pure ET (Figure 1a). It is possible that due to the fact that not the total part of ET is in the air phase the h value for ET equal 4.6 Å should be slightly smaller. 

The problem of $\Gamma^{tot}$ determination based on the $\Gamma^{GA}$ values can be solved in another way. Taking into account the limiting contactable area of water $\left(A_{W}^{0}\right)$ and ET $\left(A_{A}^{0}\right)$ molecules as well as the number of water $\left(\omega \right)$ and ET $\left(n_{A}\right)$ moles in 1 dm^3^, it is possible to calculate the surface area occupied by the water and ET molecules covering the surface by monolayer. Next, it is possible to determine the two dimensional concentration of ET in the monolayer ($\Gamma^{s}$) corresponding to its concentration in the bulk phase based on the equation:(11)$$\Gamma^{s}=\frac{n_{A}}{N({\omega A}_{W}^{0}+n_{A}A_{A}^{0})}$$Introducing this equation to Equation (10) instead of C×h one obtains:(12)$$\Gamma^{tot}=\Gamma^{GA}+\frac{n_{A}}{N(\omega A_{W}^{0}+n_{A}A_{A}^{0})}$$

As follows from Figure 1a the ET total concentration at the solution-air interface determined from Equation (12) is quite real. This equation can be also applied for the aqueous solution of surfactants if instead of $\Gamma^{GA}$ Γ is used. Then:(13)$$\Gamma^{tot}=\Gamma +\frac{n_{S}}{N(\omega A_{W}^{0}+n_{S}A_{S}^{0})}$$
where the index *S* refers to the surfactants.

The values of $\Gamma^{tot}$ calculated for RL and TX165 are not much higher than those of Γ (Figure 1b,c), which confirms that the Gibbs surface excess concentration of the surfactants is practically equal to the total one. The total concentration of the surfactant and also ET can be determined using the Frumkin equation which has the form:(14)$$\gamma_{W}-\gamma_{LV}=-RT\Gamma^{max}ln\left(1-\frac{\Gamma }{\Gamma^{max}}\right)$$
where $\gamma_{W}$ is the water surface tension. 

It appeared that the isotherm of the ethanol concentration in the surface layer calculated from Equation (14) is similar to that determined based on Equation (12) (Figure 1a). There is also agreement between the isotherms of RL and ET concentrations in the surface region determined from Equations (13) and (14) (Figure 1b,c). For RL and TX165 there is also agreement between the Γ values calculated from Equations (6) and (14).

Knowing the total two dimensional concentration of ET, RL and TX165 it is possible to determine the fraction of surface occupied by their molecules ($X^{S}$). The fraction of the surface area occupied by the ET, RL and TX165 molecules can be established from the following expression:(15)$$X^{S}=\Gamma^{s}NA_{S}^{0}$$

In fact, $X^{S}$ differs from the surfactant molar fraction ($X_{M}^{s}=\frac{\Gamma^{S}}{\Gamma^{W}+\Gamma^{S}}$), which can be determined from the following equation:(16)$${\Gamma^{W}NA_{W}^{0}+\Gamma }^{s}NA_{S}^{0}=1$$

The fractions $X^{S}$ and $X_{M}^{S}$ are applied for determination of the chemical potential in the surface region. In the Langmuir isotherm adsorption equation the value of $X^{S}$ is associated with the constant *a* [1]. Using the $X^{S}$ values the chemical potential can be defined as:(17)$$\mu =\mu^{0}+RTlnX^{S}$$

In the equilibrium state the chemical potential of a given compound in the surface region is equal to that in the bulk phase. Based on Equations (4) and (17) one can obtain:(18)$$\frac{a^{*}}{X^{S}}=exp\left(\frac{\Delta G_{ads}^{0}}{RT}\right)$$
where $∆G_{ads}^{0}$ is the standard Gibbs free energy of adsorption.

As mentioned above at small surfactants concentration Equation (18) assumes the form:(19)$$\frac{C}{X^{S}}=\omega exp\left(\frac{\Delta G_{ads}^{0}}{RT}\right)$$
where $\frac{C}{X^{S}}=$ a (a is the adsorption constant).

In order to examine in which concentration range of ET, RL and TX165 in their aqueous solution the constant a has the same value, the $\frac{C}{X^{S}}$ values were calculated from Equation (15) using their total concentration determined from Equation (12) as well as from the Frumkin equation (Equation (14)). In the case of TX165 the contactable area of its molecules at the perpendicular and parallel orientations was calculated. The values of $\frac{a^{0}}{X^{S}}$ were also determined for ET. 

It proved that for ET the values of $\frac{C}{X^{S}}$ and $\frac{a^{0}}{X^{S}}$ are not constant and depend on C (Appendix A) regardless of whether they were calculated using the values $X^{S}$ determined based on the Frumkin isotherm or the equation proposed by us (Equation (12)). This confirms that probably only in the range of ET concentration from 0 to 0.01 mol/dm^3^ the values of $\frac{C}{X^{S}}$ can be constant. Unfortunately, in this range of ET C it is difficult to measure reliable values of the surface tension. It should be mentioned that above C = 1 mol/dm^3^ the changes of $\frac{C}{X^{S}}$ and $\frac{a^{0}}{X^{S}}$ a as a function of C are almost linear. From these dependences one can deduce that $f^{0}$ is not equal to unity and increases with the increasing C. It is interesting that the ET concentration in the surface region determined from the Frumkin equation (Equation (14)) in the range of *C* in the bulk phase from zero to the value corresponding to the maximal Gibbs surface excess concentration at the solution-air interface fulfils the linear form of the Langmuir equation [1]:(20)$$\frac{C}{\Gamma }=\frac{C}{\Gamma^{max}}+\frac{a}{\Gamma^{max}}$$

The value of the constant a determined from Equation (20) is similar to that of $\frac{C}{X^{S}}$ at the low ET concentration in the aqueous solution. As the matter of fact, in a such case the $∆G_{ads}^{0}$ of ET calculated from Equation (19) based on $\frac{C}{X^{S}}$ and a is similar (Appendix A). It should be mentioned that the $\frac{a^{0}}{X^{S}}$ values are close to those determined from Equation (2) in the whole range of ET concentration.

For the aqueous solution of RL and TX165 the values of Γ obtained both from the Frumkin equation as well as our equation fulfills Equation (20). The constants a obtained from Equation (20) are similar to $\frac{C}{X^{S}}$ which are constant in the range of TX165 and RL concentration in which they are present in the bulk phase in the monomeric form (Appendix A). In the case of TX165 $X^{S}$ was calculated using the limiting contactable area of TX165 molecule at its perpendicular and parallel orientations of TX165 molecule tail. From the calculation of $X^{S}$ using the contactable area of TX165 at the perpendicular orientation it appeared that the $X^{S}$ maximal value is smaller than 0.5 and the area occupied by the water molecules is larger than the contactable area of TX165 molecule tail.

There are different possibilities of TX165 molecule orientation at the water-air interface. As it was mentioned above the hydrophilic oxyethylene groups are strongly hydrated. Moreover, it is possible that H_3_O^+^ ions are joined with the oxyethylene group [44,45]. For this reason the hydrophilic long part of TX165 molecule cannot be oriented perpendicularly towards the water-air interface at the perpendicular orientation of tail. This way of orientation increases the area occupied by one TX165 molecule at the interface. This is also another way of TX165 molecule orientation. The tail is oriented parallel to the water-air interface and the head perpendicularly. This way of TX165 molecule orientation also increase its contactable area. However, the two ways of TX165 molecules orientation give almost the same values of $\frac{C}{X^{S}}$ which are close to that of a determined from Equation (20) as well as from the Szyszkowski equation (Appendix A). The same dependences as for TX165 take place in the case of RL (Appendix A) (Appendix A). However, the maximal fraction of the surface occupied by the RL molecules is close to its perpendicular orientation. Indeed, the values of $∆G_{ads}^{0}$ calculated from Equation (19) using the values of constant a determined from the linear form of the Langmuir equation, Szyszkowski equation and calculated from $\frac{C}{X^{S}}$ are similar for a given surfactant.

### 2.2. Surface Behaviour of ET + RL + TX165 Mixtures

Behaviour of the RL + ET + TX165 mixture at the solution-air interface was considered based on the surface tension measurements of the aqueous solution of this mixture at the constant concentration of the RL mixture with ET and the variable TX165 concentration from zero to that higher than its CMC. For better understanding the behaviour of this mixture at the solution-air interface there were considered the surface tension isotherms of the binary mixtures of the components present in the ternary ones at this interface. For these considerations the surface tension isotherms of the aqueous solutions of TX165 mixtures with RL as well as RL with ET were taken from the literature [26,46]. The surface tension isotherms of the aqueous solution of the TX165 mixtures with ET were determined by the surface tension measurements as a function of TX165 concentration at the constant ET concentration (Figure 2 and Appendix A). The constant concentration of RL and ET mixture applied for measurements of the surface tension of solution of RL + ET + TX165 mixture at the TX165 variable concentration was selected based on the RL and ET individual concentration at the solution-air interface. The chosen ET concentrations were equal to 1.07, 3.74, 6.69 and 10.27 mol/dm^3^. These concentrations in the bulk phase were close to the ET concentration corresponding to its unsaturated monolayer at the solution-air interface $\left(C_{ET}^{unsat}\right)$ the maximum Gibbs surface excess concentration $\left(C_{ET}^{max}\right)$, critical aggregation concentration (CAC) and to that higher than CAC, respectively.

In the case of RL the constant concentration were equal to 0.01 (1.98 × 10^−8^ mol/dm^3^), 0.5 (9.92 × 10^−7^ mol/dm^3^), 5 (9.92 × 10^−6^ mol/dm^3^) and 20 mg/dm^3^ (3.96 × 10^−5^ mol/dm^3^). The RL concentrations in the bulk phase equal to 0.01, 0.5 and 5 mg/dm^3^ corresponded to the unsaturated monolayer at the water-air interface $\left(C_{RL}^{unsat}\right)$, the first concentration at which the saturated RL layer was formed $\left(C_{RL}^{f,sat}\right)$ and to that smaller than CMC but larger than $C_{RL}^{f,sat}$, respectively. The RL concentration equal 20 mg/dm^3^ is close to its CMC [47,48].

As mentioned above, the surface tension of the TX165 and RL tails does not differ much from the LW component of the ET surface tension. However, their ability to reduce the water surface tension by the formed adsorption layer is different. Adsorption tendency of ET, RL and TX165 depends, among others, on the number of water molecules that can contact with them and the energy effect of this contact. The ET molecules in the aqueous environment can contact with each other as well as with water ones. Due to the small difference between the surfaces of water molecules and ET, the energy effect of their contact is not great, as evidenced by the small absolute value of the Gibbs adsorption free energy. It is different in the case of the behaviour of water molecules with those of RL and TX165. 

The changes of energy of the aqueous solution of RL and TX165 result from the orientation of water molecules relative to the tail and head of surfactant molecules. The orientation of the water molecules around the head of the RL and TX165 molecules causes a decrease in the solution energy, which depends on the number and strength of hydrogen bonds. The number of water molecules hydrogenbound to the head of the RL molecule are much smaller than that of water molecules connected to the TX165 head as mentioned earlier. The number of water molecules that can contact with the tail of the RL molecule are also much smaller than in the case of the TX165 molecule. However, as mentioned above, the ratio of water molecules surrounding the head of the TX165 molecule to those surrounding the tail is much higher than in the case of RL Probably for this reason, the effect of the water surface tension reduction by adsorption of RL molecules at the water-air interface is greater than that of TX165. Moreover, RL is a weak organic acid and repulsive electrostatic interactions can play a role in the adsorption of its molecules at the water-air interface. However, due to the strong bond between the oxyethylene group and H_3_O^+^, the head of the TX165 molecule can become ionic. In this case weak repulsive intermolecular interactions can occur.

In the case of the RL and TX165 mixture synergism in the reduction of water surface tension was present but it was smaller than expected [26]. The mutual influence of RL with ET as well as TX165 with ET mixtures on the reduction of the water surface tension is different from that of the mixture of RL with TX165 (Figure 2 and Appendix A). If ET is treated as co-solvent, then there is formed the aqueous and ET solution of RL or TX165 and the RL mixture with TX165. In such case the RL or TX165 molecules adsorb at the water + ET—air interface and the mixture of RL with TX165 adsorbs at the water-air one. 

The tendency to adsorb RL and TX165 molecules at the water + ET—air interface depends on the competition of water and ET molecules in the solution bulk phase for the contact with the tail and head of RL or TX165 molecules [26]. If the ET molecules substitute for the water molecules surrounding only the head of RL or TX165 molecules, then the tendency to adsorb at the interface of RL or TX 165 should increase. In the case the water molecules surroundings tail of RL or TX165 ones are displaced by ET molecules it should decrease. Due to the fact that more water molecules surround the head of TX165 molecule than RL, the ET has a greater effect on the adsorption of TX165 than on RL. This conclusion is confirmed by the surface tension isotherms of the mixtures of RL with ET and TX165 with ET at the ET concentrations equal to $C_{ET}^{unsat}$ (Figure 2 and Appendix A). However, at the concentration of ET higher than that of CAC [30], the surface tension of the aqueous solutions of RL and ET as well as ET and TX165 mixtures is close to that of the ET aqueous solution (Figure 2, Appendix A). In this case, it is difficult to assess the mutual influence of ET and RL as well as of ET and TX165 on the water surface tension reduction. However, the adsorption of RL and TX165 at the water + ET-air interface is not excluded due to the surface tension of RL and TX165 tail similar to that of ET [26,30]. In the case of the aqueous solutions of the RL + ET + TX165 mixture, with the constant concentration of RL + ET mixture equal $C_{RL}^{unsat}$ and $C_{ET}^{unsat}$, respectively, and the variable TX165 concentration, ET has a greater effect on the water surface tension reduction than RL (Figure 3, Figure 4, Figure 5 and Figure 6). For this arrangement, the minimum surface tension is smaller than that of individual TX165 [49]. 

The surface tension isotherms for this system can be described not only by the second-order exponential function but also by the Szyszkowski equation (Figure 3, Figure 4, Figure 5 and Figure 6). 

With the increase of ET concentration to or above CAC and RL to the concentration close to its CMC, the surface tension of the RL + ET + TX165 aqueous solution does not change much as a function of the TX165 concentration and its values do not differ much from the surface tension of the individual ET aqueous solution (Figure 3, Figure 4, Figure 5, Figure 6 and Appendix A) [30]. However, the values of the surface tension of aqueous solutions of the RL + ET + TX165 mixture at the constant ET concentration equal to 10.27 mol/dm^3^ similar to the surface tension of an individual ET aqueous solution does not prove that there is no adsorption of RL and TX165 at the solution-air interface. As mentioned above the surface tension of RL and TX165 tails is close to that of ET. Therefore, it is difficult to determine directly the lack of adsorption of RL and TX165 at the high concentration of ET based only on the surface tension. Our previous studies [26,32] proved that the composition of the adsorption monolayer at the first approximation, can be predicted from the surface tension isotherms of aqueous solutions of the individual mixture components. Thus, it is possible to explain the presence in the adsorption monolayer of not only ET molecules but also RL as well as TX165 at the ET concentration at which the surface tension of the mixture solution is close to that of the ET itself.

According to this suggestion the relative molar fraction of particular components of the mixture $\left(X^{R}\right)$ can be expressed by $X_{ET}^{R}=\frac{\pi_{ET}}{\pi_{ET}+\pi_{RL}+\pi_{TX165}}$, $X_{RL}^{R}=\frac{\pi_{RL}}{\pi_{ET}+\pi_{RL}+\pi_{TX165}}$ and $X_{TX165}^{R}=\frac{\pi_{TX165}}{\pi_{ET}+\pi_{RL}+\pi_{TX165}}$, respectively.

Based on the relative molar fraction of ET, RL and TX165 in the mixed monolayer one can calculate their contribution to the water surface tension reduction from the following expression:(21)$$\gamma_{W}-\gamma_{LV}=\pi =\pi X_{ET}^{R}+\pi X_{RL}^{R}+\pi X_{TX165}^{R}$$

Taking into account the contribution of particular components to the reduction of the water surface tension it was possible to determine the concentration of all components of the binary and ternary mixtures in the mixed surface layer using the Frumkin equation (Equation (14) (Appendix A)). In the case of TX165 it was also possible to calculate its surface excess concentration from the Gibbs isotherm equation (Equation (6)). It proved that the Gibbs surface excess concentration of TX165 in the systems in which it was possible to determine is close to that calculated from the Frumkin equation. As follows Equation (21) is useful for determination of the concentration of all components of the binary and ternary mixtures at the solution-air interface. The calculations indicate that the same values of the surface tension of the aqueous solution of the binary and ternary mixtures as that of the aqueous solution of individual ET does not prove the absence of RL and TX165 in the surface region. 

The sum of the Frumkin isotherms adsorption of the binary and ternary mixtures suggests that the packing of mixed monolayers is greater than that of single one of particular components of the mixture or that the TX165 or RL or RL + TX165 mixtures are adsorbed not at the water-air but at the water + ET-air interface (Appendix A). From the Frumkin isotherms of the surface concentration of the particular components of the mixed monolayer it was also possible to determine the adsorption constants for all studied systems as well as the standard Gibbs free energy of adsorption using Equations (15) and (19). Such calculations were made both for the binary and ternary systems (Appendix A). The adsorption constants were compared to those calculated from the Langmuir linear equation as well as from the Szyszkowski equation (Appendix A). 

The $\frac{C}{X^{S}}$ values for TX165 both in the binary and ternary mixtures are constant only in the range of its low concentration and are almost the same independently of the constant concentration of ET and RL (Appendix A). However, the $\frac{C}{X^{S}}$ values for RL and ET practically do not depend on the TX165 concentration but on their constant concentration values to a large extent and in the case of ET and RL mixture also on its composition (Appendix A). The values of $\frac{C}{X^{S}}$ for TX165, RL and ET differ a little from the constant adsorption obtained from the Szyszkowski and linear Langmuir equations. Thus, in the case of TX165 it takes place only in the range of its low concentration. However, it was not possible to determine the constant adsorption for all studied systems using the linear Langmuir and Szyszkowski equations.

Moreover it should be emphasized that in some cases both for the binary and ternary mixtures the total values of $X^{S}$ are greater than unity. This confirms the above mentioned conclusion that the layer of RL or TX165 or RL + TX165 can be formed at the water + ET mixed solvent-air interface. Thus, the surface region can be treated as the sum of two monolayers, the ET and mixed RL + TX165 one. Such ET, RL and TX165 behaviour explains why the sum of the surface fraction occupied by particular components of the ternary system is greater than zero.

Taking into account the adsorption constants the standard Gibbs free energy of adsorption of ET, RL and TX165 was calculated from Equation (19) (Appendix A) (Appendix A). As follows from the calculations both RL and ET influence on TX165 tendency to adsorb only in a small extent and the absolute value of $∆G_{ads}^{0}$ both for the binary and ternary systems is close to that of individual TX165 at its low concentration (Appendix A) (Appendix A). In the case of RL and ET the $∆G_{ads}^{0}$ values depend on their individual mutual constant concentrations as well as that of TX165 (Appendix A).

## 3. Materials and Methods

### 3.1. Materials

Ethanol (ET) (96% pure) was purchased from POCH (Gliwice, Poland). Before the solutions preparation ET was purified by the method described earlier [30,46]. Triton X-165 (TX165) ((*p*-(1,1,3,3-tetramethylbutyl)-phenoxypolyoxyethylene glycol) of the purity greater than 99% was purchased from Fluka and R-95 Rhamnolipid (95%) was from Sigma-Aldrich (St. Louis, MO, USA). For the solution preparation TX165 and RL were used without further purification.

For the surface tension measurements there were prepared four series of the aqueous solution of TX165 mixtures with ET and sixteen series for the aqueous solution of the RL + ET + TX165 mixture. The series of the aqueous solution of TX165 mixtures with ET had the constant concentration of ET equal 1.07, 3.74, 6.69 or 10.27 mol/dm^3^ and the variable concentration of TX165 from 0 to 4 × 10^−^^3^ mol/dm^3^. The series of the aqueous solutions of RL + ET +TX165 mixture included the constant concentration of the ET + RL sum and the variable concentration of TX165 as mentioned above. The constant sum concentration of ET + RL was prepared from all possible combinations of ET at the concentrations equal 1.07, 3.74, 6.69 and 10.27 mol/dm^3^ and RL at the concentrations equal 0.01 (1.98 × 10^−^^8^ mol/dm^3^), 0.5 (9.92 × 10^−^^7^ mol/dm^3^), 5 (9.92 × 10^−^^7^ mol/dm^3^) and 20 mg/dm^3^ (3.96 × 10^−^^5^ mol/dm^3^), respectively.

### 3.2. Methods

The surface tension ($\gamma_{LV}$) of the aqueous solution of the ET + TX165 mixture at the constant concentration of ET and variable concentration of TX165 as well as the aqueous solution of ET + RL + TX165 mixture and TX165 variable RL + ET + TX165 was measured by the Krüss K9 tensiometer (Krüss, Hamburg, Germany) according to the platinum ring detachment method (du Nouy’s method) at 293 K. Before the surface tension measurements of the studied aqueous solutions the tensiometer was calibrated based on the measurements of water and methanol surface tension. For each concentration of the TX165 at each series of the solutions, the surface tension measurements were repeated at least ten times. The standard deviation was ±0.1 mN/m and the uncertainty of the surface tension measurements was in the range from 0.3% to 0.7%.

## 4. Conclusions

Based on the obtained results and their thermodynamic analysis, a number of conclusions can be drawn.

The surface tension isotherm of the aqueous solution of particular components of the ET + RL + TX 165 mixture can be described by the exponential function of the second order. However, taking into account the exponential function equation and Gibbs isotherm of the surface excess concentration one it is impossible to obtain the real surface excess concentration in whole range of its concentrations in the bulk phase. The more real seems to be the Guggenheim-Adam isotherm of excess concentration of ET at the water-air interface in the whole range of its concentrations in the bulk phase.

We have proposed a simple equation for calculation of the total ET concentration at the water-air interface, taking into account the isotherm of the excess ET concentration calculated from the Guggenheim-Adam equation.

The total concentration of ET calculated by our equation is in accordance with that obtained from the Frumkin equation. On the basis of our equation it was proved that the Gibbs surface excess concentration of RL and ET at the water-air interface is practically equal to the total one.

The total concentration allows to determine the fraction of the surface area occupied by ET, RL and TX165 at the water-air interface in their individual solutions as well as in the solution of the binary and ternary mixtures of these compounds.

The ratio of the mole fraction of the ET, RL and TX165 to that of the surface area occupied by them is close to the adsorption constant determined from the linear Langmuir and Szyszkowski equations in a range of ET, RL and TX165 concentrations.

Based on the surface fraction occupied by ET, RL and TX165 it was deduced that RL and TX165 can be adsorbed at the mixed solvent (water + ET) –air interface.

It proved that adsorption of RL and TX165 can take place even when the surface tension of the aqueous solution of the binary or ternary systems including ET is close to that of ET individual solution.

At its low concentration ET influences on the standard Gibbs free energy of RL and TX165 adsorption to a small extent.

## Figures and Tables

**Figure 1 molecules-28-04987-f001:**
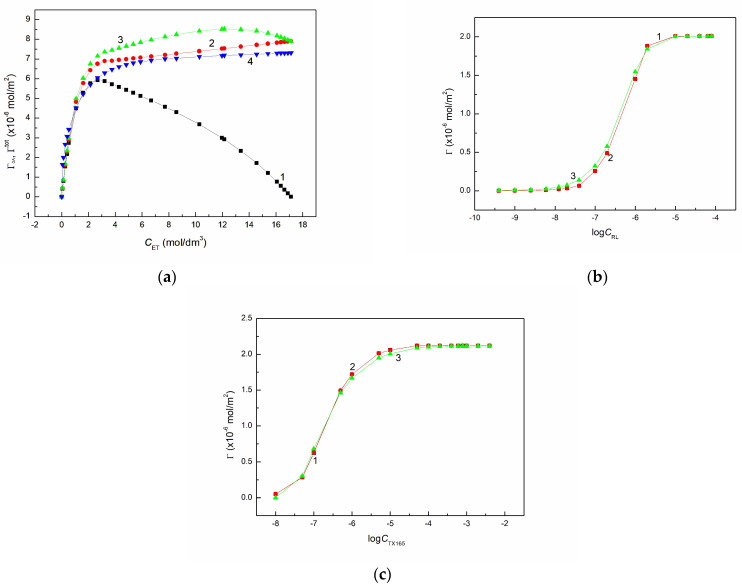
A plot of the ET Guggenheim-Adam excess concentration at the water-air interface ($\Gamma^{GA}$) (curve 1—correspond to the values calculated from Equation (8)) as well a plot of the ET total concentration ($\Gamma^{tot}$) (curves 2–4 correspond to the values calculated from Equations (10), (12) and (14), respectively) vs. its concentration ($C_{ET}$) (**a**) and a plot of the RL (**b**) and TX165 (**c**) Gibbs surface excess concentration (Γ) (curve 1 correspond to the values calculated from Equation (6)) as well as its total concentration $(\Gamma^{tot}$) (curves 2 and 3 correspond the values calculated from Equations (12) and (14) vs. the logarithm of the surfactant concentration $\left(logC\right)$.

**Figure 2 molecules-28-04987-f002:**
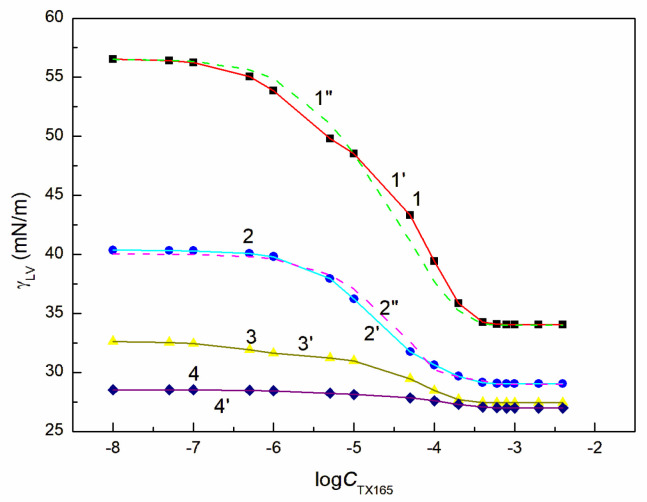
A plot of the surface tension $\left(\gamma_{LV}\right)$ of the aqueous solution of the TX165 + ET mixture at the constant ET concentration equal to 1.07 mol/dm^3^ (points 1, curves 1′ and 1″), 3.74 mol/dm^3^ (points 2, curves 2′ and 2″), 6.69 mol/dm^3^ (points 3, curve 3′) and 10.27 mol/dm^3^ (points 4, curve 4′) vs. the logarithm of the TX165 concentration $\left(C_{TX165}\right)$. Points 1–4 correspond to the measured values, curves 1′–4′ and curves 1″, 2″ correspond to the values calculated from Equations (1) and (2), respectively.

**Figure 3 molecules-28-04987-f003:**
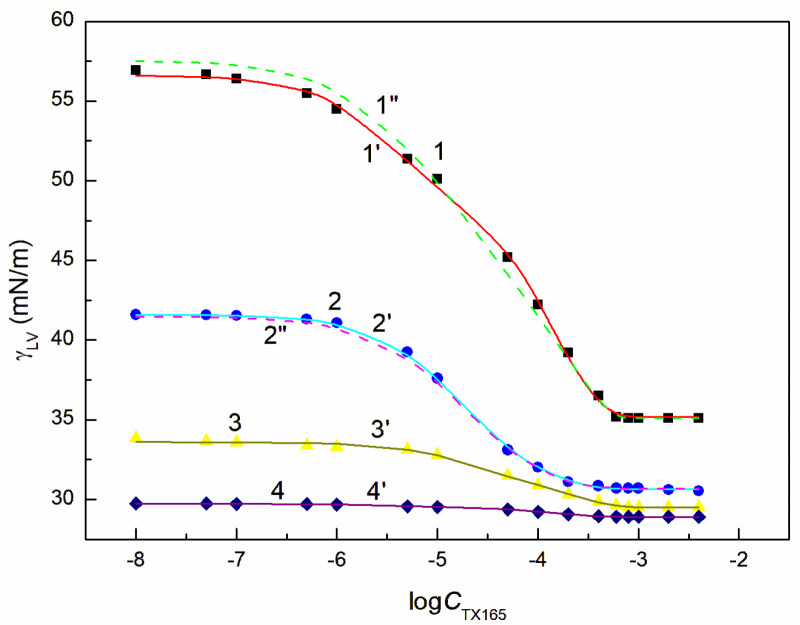
A plot of the surface tension $\left(\gamma_{LV}\right)$ of the aqueous solution of the RL + ET + TX165 mixture at the constant RL concentration equal to 0.01 mg/dm^3^ and ET concentration equal to 1.07 mol/dm^3^ (points 1, curves 1′ and 1″), 3.74 mol/dm^3^ (points 2, curves 2′ and 2″), 6.69 mol/dm^3^ (points 3, curve 3′) and 10.27 mol/dm^3^ (points 4, curve 4′) vs. the logarithm of the TX165 concentration $\left(C_{TX165}\right)$. Points 1–4 correspond to the measured values, curves 1′–4′ and curves 1″, 2″ correspond to the values calculated from Equations (1) and (2), respectively.

**Figure 4 molecules-28-04987-f004:**
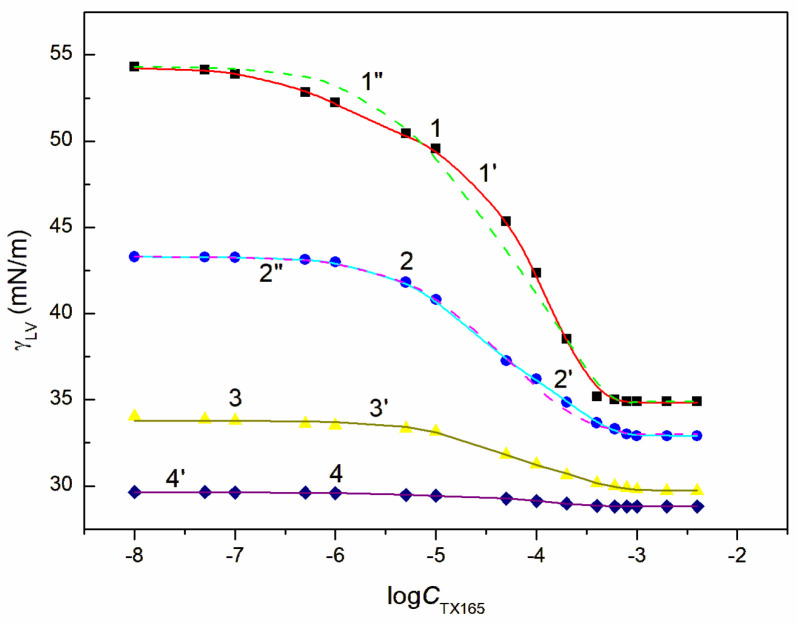
A plot of the surface tension $\left(\gamma_{LV}\right)$ of the aqueous solution of the RL + ET + TX165 mixture at the constant RL concentration equal to 0.5 mg/dm^3^ and ET concentration equal to 1.07 mol/dm^3^ (points 1, curves 1′ and 1″), 3.74 mol/dm^3^ (points 2, curves 2′ and 2″), 6.69 mol/dm^3^ (points 3, curve 3′) and 10.27 mol/dm^3^ (points 4, curve 4′) vs. the logarithm of the TX165 concentration $\left(C_{TX165}\right)$. Points 1–4 correspond to the measured values, curves 1′–4′ and curves 1″, 2″ correspond to the values calculated from Equations (1) and (2), respectively.

**Figure 5 molecules-28-04987-f005:**
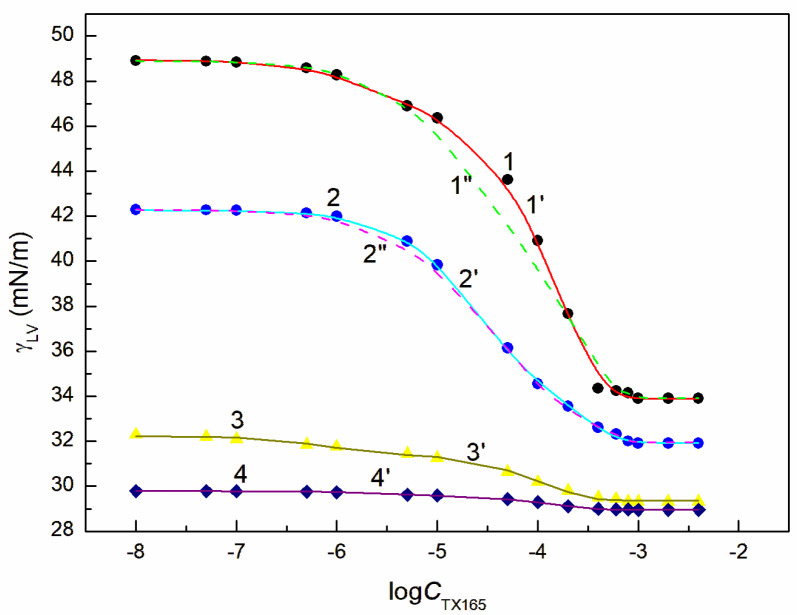
A plot of the surface tension $\left(\gamma_{LV}\right)$ of the aqueous solution of the RL + ET + TX165 mixture at the constant RL concentration equal to 5 mg/dm^3^ and ET concentration equal to 1.07 mol/dm^3^ (points 1, curves 1′ and 1″), 3.74 mol/dm^3^ (points 2, curves 2′ and 2″), 6.69 mol/dm^3^ (points 3, curve 3′) and 10.27 mol/dm^3^ (points 4, curve 4′) vs. the logarithm of the TX165 concentration $\left(C_{TX165}\right)$. Points 1–4 correspond to the measured values, curves 1′–4′ and curves 1″, 2″ correspond to the values calculated from Equations (1) and (2), respectively.

**Figure 6 molecules-28-04987-f006:**
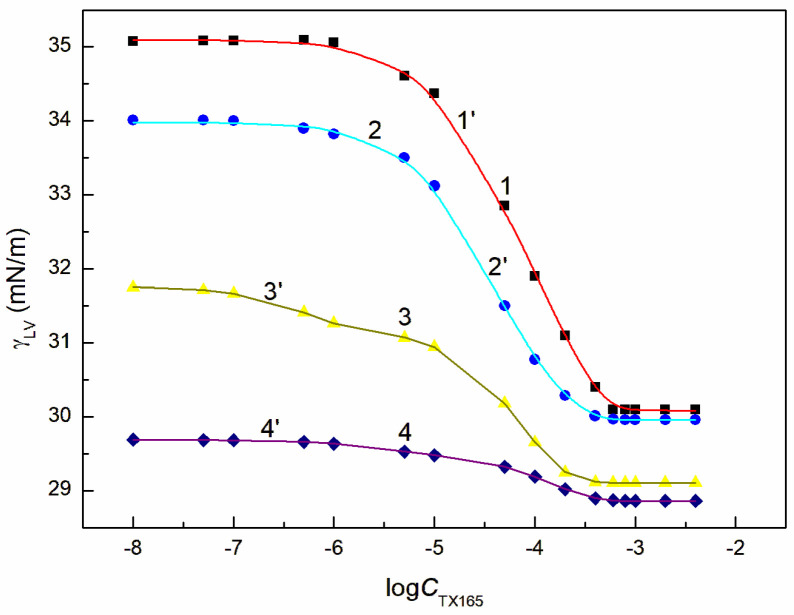
A plot of the surface tension $\left(\gamma_{LV}\right)$ of the aqueous solution of the RL + ET + TX165 mixture at the constant RL concentration equal to 20 mg/dm^3^ and ET concentration equal to 1.07 mol/dm^3^ (points 1and curve 1′), 3.74 mol/dm^3^ (points 2and curve 2′), 6.69 mol/dm^3^ (points 3 curve 3′) and 10.27 mol/dm^3^ (points 4, curve 4′) vs. the logarithm of the TX165 concentration $\left(C_{TX165}\right)$. Points 1–4 correspond to the measured values, curves 1′–4′ correspond to the values calculated from Equation (1).

**Table 1 molecules-28-04987-t001:** Components and parameters of the ET, TX165 and RL surface tension ($\gamma_{LV}$) at 293 K, the maximal concentration at the water-air interface ($\Gamma^{max}$), limiting concentration at the water-air interface ($\Gamma^{0}$) and limiting area occupied by one water, TX165, ET and RL molecule ($A_{0}$).

Substance	$\gamma_{LV}^{LW}$ [mN/m]	$\gamma_{LV}^{+}$ [mN/m]	$\gamma_{LV}^{-}$ [mN/m]	$\gamma_{LV}^{AB}$ [mN/m]	$\gamma_{LV}$ [mN/m]	$A_{0}$ [Å^2^]	$\Gamma^{max}$ [×10^−^^6^ mol/m^2^]	$\Gamma^{0}$ [×10^−^^6^ mol/m^2^]	Ref.
Water from $\gamma_{WH}$	21.80	25.60	25.50	51.00	72.80	10.00	16.60	16.60	[39]
Water from θ	26.85	22.975	22.975	45.95	72.80	-	-	-	[40]
ET	21.40	0.09	9.00	1.80	24.20	21.00	7.91	7.91	[30]
TX165 tail	22.00	-	-	-	22.00	35.70	2.12	4.65	[26]
TX165 head	27.70	0.33	50.20	8.14	35.84	-	-	-	[26]
RL tail	21.80	-	-	-	21.80	69.09	2.01	2.403	[26]
RL head	35.38	0.04	56.74	3.01	38.39	-	-	-	[26]

## Data Availability

The data presented in this study are available in Appendix A.

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
