# Peer review of "Thermodynamic Characterization of Rhamnolipid, Triton X-165 and Ethanol as well as Their Mixture Behaviour at the Water-Air Interface"

_molecules, 2023, doi:10.3390/molecules28134987_

Round 1
Reviewer 1 Report
Very interesting work, contributing new data to the current state of knowledge. Minor editorial errors, not affecting the evaluation of the quality of the manuscript.
The manuscript is directed to the study of adsorption properties of aqueous solutions of mixtures containing a biosurfactant and a non-ionic surfactant or short-chain alcohol. Biosurfactants, which are amphiphilic organic compounds of natural origin, produced by microorganisms, as well as secreted by some multicellular eukaryotic organisms, were chosen as the object of study. Although biosurfactants have very high application potential, their widespread use in industry has some limitations. First is the cost. Biosurfactants are often more expensive to produce than traditional surfactants. Compared to traditional surfactants, biosurfactants are also often less efficient, meaning that more product is required to achieve the same effect. On the other hand, biosurfactants are seen as an alternative to traditional surfactants, which is important in the context of sustainable development and efforts to minimize the negative environmental impact of industry. The use of biosurfactants in mixtures with classical surfactants seems to be the answer to the search for a compromise on the issues presented. Comprehensive analysis of the properties of mixtures based on the physicochemical and surface properties of the individual components, and especially the ability to predict these properties, are key to application solutions.
The scope of the reviewed paper fits perfectly into the research area concerning the physicochemical and surface properties of amphiphilic compounds. According to the literature, the reviewed work contains elements of scientific novelty. In the literature there is a lack of works on the comprehensive analysis of the properties of mixtures containing a biosurfactant and a classical surfactant, without or with organic additives, based on the physicochemical properties of the individual components, and especially the possibility of predicting these properties. The authors' undertaking of research on the mutual effects of rhamnolipid and surfactant, Triton X-165, and ethanol on adsorption to some extent fills this gap.
The authors cite relevant literature, present the corresponding conclusions. In the future it is worth considering the use of another non-ionic surfactant compound, there is a wide range outside the Triton series (more environmentally friendly surfactant).
Author Response
Reply to the Reviewer 1 comments
Thank you very much for the time invested in the revision of our manuscript and for its favourable evaluation.
Very interesting work, contributing new data to the current state of knowledge. Minor editorial errors, not affecting the evaluation of the quality of the manuscript.
The manuscript is directed to the study of adsorption properties of aqueous solutions of mixtures containing a biosurfactant and a non-ionic surfactant or short-chain alcohol. Biosurfactants, which are amphiphilic organic compounds of natural origin, produced by microorganisms, as well as secreted by some multicellular eukaryotic organisms, were chosen as the object of study. Although biosurfactants have very high application potential, their widespread use in industry has some limitations. First is the cost. Biosurfactants are often more expensive to produce than traditional surfactants. Compared to traditional surfactants, biosurfactants are also often less efficient, meaning that more product is required to achieve the same effect. On the other hand, biosurfactants are seen as an alternative to traditional surfactants, which is important in the context of sustainable development and efforts to minimize the negative environmental impact of industry. The use of biosurfactants in mixtures with classical surfactants seems to be the answer to the search for a compromise on the issues presented. Comprehensive analysis of the properties of mixtures based on the physicochemical and surface properties of the individual components, and especially the ability to predict these properties, are key to application solutions.
The scope of the reviewed paper fits perfectly into the research area concerning the physicochemical and surface properties of amphiphilic compounds. According to the literature, the reviewed work contains elements of scientific novelty. In the literature there is a lack of works on the comprehensive analysis of the properties of mixtures containing a biosurfactant and a classical surfactant, without or with organic additives, based on the physicochemical properties of the individual components, and especially the possibility of predicting these properties. The authors' undertaking of research on the mutual effects of rhamnolipid and surfactant, Triton X-165, and ethanol on adsorption to some extent fills this gap.
The authors cite relevant literature, present the corresponding conclusions. In the future it is worth considering the use of another non-ionic surfactant compound, there is a wide range outside the Triton series (more environmentally friendly surfactant).
Thank you very much again for your valuable comments and favourable evaluation of our manuscript.
Reviewer 2 Report
The work is clearly performed in a professional and serious manner, the controls placed on the experiments are sufficient to trust the results, the interpretations of the data are in general appropriate, the list of references is broad, appropriate and up-to-date. The work is recommended for publication following consideration of the authors of only a few very minor points:
1. The format in the paper needs to be carefully revised, for example: p.15, line 547: “[46,30?]”, "?" cannot appear here. Similar issues should be carefully revised throughout the paper.
2. The unit of ET is mol/dm3, the unit of RL and TX165 is mg/dm3. Why cannot the units be unified? They all become mg/dm3.
1. The format in the paper needs to be carefully revised, for example: p.15, line 547: “[46,30?]”, "?" cannot appear here. Similar issues should be carefully revised throughout the paper.
2. The unit of ET is mol/dm3, the unit of RL and TX165 is mg/dm3. Why cannot the units be unified? They all become mg/dm3.
Author Response
Replies to the Reviewer 2 comments
Thank you very much for the time invested in the revision of our manuscript and for the attempts you have made to improve it. The manuscript was revised based on your remarks and suggestions. Our reply to your comments is as follows:
The work is clearly performed in a professional and serious manner, the controls placed on the experiments are sufficient to trust the results, the interpretations of the data are in general appropriate, the list of references is broad, appropriate and up-to-date. The work is recommended for publication following consideration of the authors of only a few very minor points:
- The format in the paper needs to be carefully revised, for example: p.15, line 547: “[46,30?]”, "?" cannot appear here. Similar issues should be carefully revised throughout the paper.
Thank you for this remark. This is our error that has been corrected in the revised version of the manuscript. As you suggested, the whole manuscript has been carefully checked with respect to the editing errors.
- The unit of ET is mol/dm3, the unit of RL and TX165 is mg/dm3. Why cannot the units be unified? They all become mg/dm3.
You are right that we use of different units to express the concentration of compounds applied in our research. In the previous studies on the mixtures of ethanol and rhamnolipid, we used molar concentrations for ethanol and mg/dm3 for rhamnolipid (J. Molecular Liquids, 2020, 308, 113080). In this paper the same ethanol concentrations and selected rhamnolipid concentrations in the range from 0 to 40 mg/dm3 were used. Taking this into account, the same units as in the previous paper were applied. We hope that this will make it easier for readers to compare our results in this paper with the previous one as well as to interpret them. We would like also to mention that in the literature for various types of biosurfactants, their concentrations are often expressed in mg/dm3. However, for comparison of the concentrations in the revised manuscript at the rhamnolipid concentration expressed in mg/dm3 in brackets we added the concentration in mole/dm3 taking into account the molar mass of the mono-rhamnolipid.